# Crosstalk between Sodium–Glucose Cotransporter Inhibitors and Sodium–Hydrogen Exchanger 1 and 3 in Cardiometabolic Diseases

**DOI:** 10.3390/ijms222312677

**Published:** 2021-11-24

**Authors:** Al-Anood Al-Shamasi, Rozina Elkaffash, Meram Mohamed, Menatallah Rayan, Dhabya Al-Khater, Alain-Pierre Gadeau, Rashid Ahmed, Anwarul Hasan, Hussein Eldassouki, Huseyin Cagatay Yalcin, Muhammad Abdul-Ghani, Fatima Mraiche

**Affiliations:** 1Department of Pharmaceutical Sciences, College of Pharmacy, QU Health, Qatar University, Doha P.O. Box 2713, Qatar; aa1506993@qu.edu.qa (A.-A.A.-S.); re1510208@qu.edu.qa (R.E.); mi1517626@qu.edu.qa (M.M.); mr1517635@qu.edu.qa (M.R.); Da1508624@qu.edu.qa (D.A.-K.); 2Biomedical and Pharmaceutical Research Unit, QU Health, Qatar University, Doha P.O. Box 2713, Qatar; 3INSERM, Biology of Cardiovascular Disease, University of Bordeaux, U1034 Pessac, France; alain.gadeau@inserm.fr; 4Department of Mechanical and Chemical Engineering, College of Engineering, Qatar University, Doha P.O. Box 2713, Qatar; hashmi133@yahoo.com (R.A.); ahasan@qu.edu.qa (A.H.); 5Biomedical Research Centre (BRC), Qatar University, Doha P.O. Box 2713, Qatar; hyalcin@qu.edu.qa; 6College of Kinesiology, University of Saskatchewan, Saskatoon, SK S7N 5B5, Canada; hue454@mail.usask.ca; 7Division of Diabetes, University of Texas Health Science Center at San Antonio, Floyd Curl Drive, San Antonio, TX 7703, USA; abdulghani@uthscsa.edu

**Keywords:** sodium–glucose cotransporter inhibitors, SGLT1, SGLT2, sodium–hydrogen exchanger, NHE1, NHE3, cardiovascular diseases, diabetes

## Abstract

Abnormality in glucose homeostasis due to hyperglycemia or insulin resistance is the hallmark of type 2 diabetes mellitus (T2DM). These metabolic abnormalities in T2DM lead to cellular dysfunction and the development of diabetic cardiomyopathy leading to heart failure. New antihyperglycemic agents including glucagon-like peptide-1 receptor agonists and the sodium–glucose cotransporter-2 inhibitors (SGLT2i) have been shown to attenuate endothelial dysfunction at the cellular level. In addition, they improved cardiovascular safety by exhibiting cardioprotective effects. The mechanism by which these drugs exert their cardioprotective effects is unknown, although recent studies have shown that cardiovascular homeostasis occurs through the interplay of the sodium–hydrogen exchangers (NHE), specifically NHE1 and NHE3, with SGLT2i. Another theoretical explanation for the cardioprotective effects of SGLT2i is through natriuresis by the kidney. This theory highlights the possible involvement of renal NHE transporters in the management of heart failure. This review outlines the possible mechanisms responsible for causing diabetic cardiomyopathy and discusses the interaction between NHE and SGLT2i in cardiovascular diseases.

## 1. Introduction

Type 2 Diabetic Melitus (T2DM) affects approximately 463 million people worldwide and future estimates suggest that 102 out of every 1000 people will be diagnosed with diabetes by 2030 [1]. Diabetes mellitus (DM) is a metabolic disorder in which the body either does not produce sufficient amounts of insulin, has impaired insulin action or a combination of the two. Type 1 Diabetes mellitus (T1DM), which constitutes ~5–10% of diabetes cases, has a high incidence in children and adolescents and is caused by the destruction of the insulin producing β-islets in the pancreas. The majority of DM cases are T2DM, which results from a combination of beta cell dysfunction and insulin resistance. The insulin resistance and the accompanied hyperinsulinemia are the early detected metabolic abnormalities in subjects destined to develop T2DM which precede the deterioration in glucose homeostasis [2,3]. Chronic diabetes without appropriate treatment causes microvascular and macrovascular complications like nephropathy, retinopathy, neuropathy, and atherosclerotic cardiovascular diseases (CVDs).

CVDs are considered to be the most common causes of morbidity and mortality in diabetic patients. In the US, CVD related death rates are 1.7 times higher among adults with DM compared to those for adults without DM, which is attributable to the increased risk of stroke, myocardial infarction (MI), and heart failure (HF) [4]. Patients with T2DM have a two-to-five-fold increased risk of HF, independent of other risk factors such as hypertension, coronary artery disease, and dyslipidemia [5,6]. A rise in glycated hemoglobin by 1% has been associated with an 8% increase in CVD risk [7]. Furthermore, the presence of T2DM worsens the prognosis of heart failure. In addition, T1DM patients have a 30% risk of HF with every 1% increase in glycated hemoglobin [8].

Hyperglycemia and insulin resistance are the major etiological factors promoting cardiomyopathy and HF in diabetic patients [9]. Ultimately, the progression of HF in DM is linked to pathological changes to the heart muscle and coronary vasculature, which eventually lead to diabetic cardiomyopathy (DCM) [10].

Recently discovered antihyperglycemic agents including glucagon-like peptide-1 receptor agonists and the sodium–glucose cotransporter-2 (SGLT2) inhibitors (SGLT2i) have shown cardioprotective effects. SGLT2i were shown to decrease the rates of HF and hospitalization from HF in several clinical trials [11]. The exact mechanism in which the new antihyperglycemic drugs exert cardioprotection is unknown, although some studies show involvement of the sodium–hydrogen exchanger (NHE) and sodium–glucose transporter (SGLT) families. This review discusses some of the mechanisms predisposing to diabetic cardiomyopathy and highlights the role of NHE and SGLT transporters in cardiovascular diseases.

## 2. Pathophysiology of Diabetic Cardiomyopathy

DCM is recognized by defects in the structure and performance of the myocardium in individuals with diabetes, independent of other cardiac risk factors. The structural abnormalities in DCM progress through three stages: an early stage characterized by diastolic dysfunction, which gradually develops to systolic dysfunction in the advanced stage, and eventually to HF in the late stage [9]. Early-stage DCM, mainly caused by hyperglycemia and insulin resistance, presents with impairment in the left ventricle (LV) diastolic filling, compensated by increased LV and atrial filling pressure and left atrial enlargement [12]. Hyperglycemia leads to the downregulation of GLUT4, impaired glycolysis, and increased free FA levels (from impaired FA metabolism). Insulin resistance results in increased lipolysis and an elevated plasma FFA concentration as well as leading to an increased influx to myocytes and the development of cardiac steatosis. The events result in high levels of reactive oxygen species (ROS), impaired calcium (Ca^2+^) homeostasis, mitochondrial dysfunction, endoplasmic reticulum stress, oxidative stress, and activation of the sympathetic nervous system, all of which promote cardiac hypertrophy, fibrosis, and cardiomyocyte apoptosis [9]. The advanced stage comprises continued cardiac injury and further stimulates the renin–angiotensin–aldosterone system (RAAS) and maladaptive immune responses that culminate in the impaired autophagy of cells [9]. Advanced stage features include LV hypertrophy and cardiac remodelling, with impaired cardiac diastolic function. Consequently, the individual may develop HF with a normal ejection fraction. In late-stage DCM, neurohumoral activation, impaired metabolism, and myocardial fibrosis weaken coronary microcirculation and the diastolic and systolic functions of the heart [10,13]. Additionally, impaired insulin signalling and oxidative stress both decrease levels of the vasodilator nitric oxide (NO) [9].

### 2.1. The Role of Fibrogenesis in DCM

The primary cause of structural abnormalities in the settings of DCM is fibrosis. Fibrosis describes the inappropriate buildup of the extracellular matrix (ECM). When the heart starts failing, it loses cardiomyocytes that are inappropriately replaced due to the limited regenerative capacity of the heart. At the same time, the ECM starts building up to offer structural and functional support to the failing heart. However, the excessive accumulation of ECM distorts cardiac function. Fibrosis increases cardiac tissue stiffness and reduces its compliance, leading to contractile dysfunction. In a healthy heart, ECM fibers are crucial in electrical transduction through the heart. Therefore, impaired cardiac ECM predisposes the heart to arrhythmias. Cardiac fibrosis impairs pumping capacity leading to systolic dysfunction, as seen in HF with a reduced ejection fraction. While in HF with a preserved ejection fraction, fibrosis-induced stiffness disrupts ventricular filling capacity [14,15]. The process of fibrosis includes the removal of the damaged ECM and its replacement. The process involves the production of collagen, glycoproteins, and proteoglycans, such as fibronectin [16].

Mechanical stress on the heart is a driver of ECM accumulation. The upregulation of the cross-linking enzymes lysyl oxidase (LOX) and LOX-like (LOXL) enzymes are a characteristic of mechanical injuries to the heart. LOX and LOXL help in collagen cross-linking. In addition, LOXL2 stimulates the production of transforming growth factor-beta (TGF-β) from cardiac fibroblasts [14]. Additionally, TGF-β is an important player that stimulates fibroblast proliferation and conversion to myofibroblasts. It also protects myofibroblasts from apoptosis through activating the pro-survival signaling of the PI3K/AKR pathway. Myofibroblasts are responsible for the production of ECM proteins. TGF-β leads to the activation of the epidermal growth factor (EGF), insulin-like growth factor-1 (IGF-1), and growth differentiation factor-11 (GDF-11), all of which increase fibrosis [17].

Transglutaminase 2 (TG-2) is another regulator of ECM remodeling. TG2 helps in producing strong collagen fibers that resist degradation. It also helps activate TGF-β from its latent complex [14].

Although fibroblasts are the main source of activated myofibroblasts, they can also originate from different types of cells. Myofibroblasts are special cells that exhibit high secretory ability and express α-smooth muscle actin (α-SMA). One suggested source of myofibroblasts is the endothelial to mesenchymal transition (EndMT). In diabetic models, endothelin-1 was secreted by EndMT, which played an important role in cardiac fibrosis [18].

DCM is not the only cardiometabolic disease that is affected with fibrosis. Interstitial fibrosis is one of the manifestations of diabetic nephropathy. In renal fibrosis, the sources of myofibroblasts include fibroblasts, pericytes, and endothelial cells. EndMT and epithelial to mesenchymal transition (EMT) have been suggested to be the progenitors of up to 50% of myofibroblasts in renal fibrosis. Due to the transition, EndMT and EMT lose their apical/basal polarity, and become less adhesive and more motile. This leads to a reduction in the peritubular capillary density and an increase in fibrosis. Renal fibrosis leads to structural abnormalities, reduced eGFR, and eventually may lead to end-stage renal disease [19]. A deficiency in FGFR1 (fibroblast growth factor receptor 1), the endothelial receptor essential for combating EndMT, has been suggested to contribute the fibrogenic effects in the kidney and hearts of diabetic mice and is being further investigated for its therapeutic application [19].

### 2.2. The Role of Endothelial Dysfunction in the Development of DCM

Studies on DCM development using animal models have implicated multiple pathophysiologic mechanisms, such as mitochondrial dysfunction, RAAS activation, Ca^2+^ homeostasis impairment, lipotoxicity, myocardial steatosis, glucose toxicity, and most recently, endothelial dysfunction. Hyperglycemia is one of the main factors triggering endothelial dysfunction by exerting several biochemical changes that damage cardiac and vascular endothelial cells. Some of these changes trigger ROS production and induce oxidative stress levels that overwhelm cells, enhance non-enzymatic glycation, activate protein kinase-C (PKC), and ameliorate the cells’ redox potential [20]. Oxidative stress promotes the formation and deposition of AGE products creating elevated interstitial collagen deposition and increased myocardial wall stiffness. If untreated, all of these DCM-related structural changes would result in HF [9].

The endothelium is a single-layer cellular lining of the whole vascular system. Endothelial cells have unique functions that are vital for cardiovascular homeostasis. For example, the endothelium functions as a semi-permeable barrier between blood and body tissues. The endothelium also controls vascular tone by secreting the vasodilators nitric oxide, prostacyclin, and endothelium-derived hyperpolarizing factors, as well as producing vasoconstrictors like endothelin-1 and thromboxane-A_2_. Endothelial dysfunction, characterized by low nitric oxide bioavailability, occurs when endothelial cells lose their barrier property and fail to balance vascular dilatory and constrictive tone, coagulation, and anticoagulation. T1DM and T2DM patients show decreased vasorelaxation by NO [20,21]. Reduced NO production is observed in diabetic experimental models [22,23], and in vitro studies with endothelial cells have shown that high glucose levels lead to less NO production [24]. Endothelial dysfunction is considered the first step in developing atherosclerotic complications in metabolic conditions such as diabetes, pre-diabetes, and obesity [25]. T1DM and T2DM patients show decreased vasorelaxation by NO [20,21]. Reduced NO production is observed in diabetic experimental models [22,23], and in vitro studies with endothelial cells have shown that high glucose levels lead to less NO production [24]. Endothelial dysfunction is considered the first step in developing atherosclerotic complications in metabolic conditions such as diabetes, pre-diabetes, and obesity [25].

Several mechanisms contribute to lower NO bioavailability during endothelial dysfunction. The production of ROS through NADPH oxidase, an electron transport chain protein, leads to oxidative stress. ROS reacts with NO to produce a cytotoxic oxidant compound called peroxynitrite. Peroxynitrite increases oxidative stress even further, which in turn lowers NO production through the uncoupling of NO synthases (NOS) and mediates low-density lipoprotein oxidation. Peroxynitrite also leads to protein dysfunction via the nitration of proteins. In insulin resistance, the PI3K/Akt pathway involved in NOS activation is inhibited, whereas endothelin-1 and adhesion molecule production pathways remain intact. Moreover, the presence of AGE products contributes to oxidative stress and leads to endothelial dysfunction. Endoplasmic reticulum stress, a pro-apoptotic pathway, is another mechanism where the pro-survival unfolded protein response becomes chronically activated [20,21,26,27,28]. The absence of endothelial glucocorticoid receptors has been demonstrated to contribute to renal fibrosis in diabetic mice, which is associated with cytokine and chemokine reprogramming [20].

Antihyperglycemic medications that target and attenuate endothelial dysfunction such as liraglutide, metformin, pioglitazone, and SGLT2i (empagliflozin, EMPA; canagliflozin, CANA; dapagliflozin, DAPA) are becoming of great interest [29,30]. In porcine coronary artery cultured endothelial cells, high glucose increased endothelial dysfunction markers, oxidative stress, and vascular cell adhesion molecule 1 (VCAM-1), and reduced NOS expression. Treatment with SGLTi exerted a protective effect and prevented endothelial dysfunction [31]. Additionally, in the obese ZSF1 rat model, systolic blood pressure (BP) was higher than the lean control group; NOS was downregulated, and the expression of VCAM-1 was increased. The chronic treatment of T2DM ZDF rats with EMPA prevented oxidative stress, signalling and inflammation, AGE products formation, and attenuated endothelial dysfunction [32]. In Apo-E^−/−^ streptozotocin (STZ)-induced diabetic mice, treatment with EMPA also attenuated endothelial dysfunction and reduced atherogenesis [33]. The EMBLEM trial included 117 patients with T2DM and concurrent CVDs, randomized into a 1:1 ratio to receive either a placebo or EMPA over 24 weeks [34]. The primary endpoint was the change in the reactive hyperemia index, an endothelial dysfunction marker, from baseline. The per-protocol analysis did not show an improvement in endothelial dysfunction. However, the study was limited by the small number of patients and unrepresentable population. The mean population systolic BP was 130 mmHg and the BMI 26.4 kg m^−2^, which are lower than expected values for diabetic patients with concurrent cardiovascular disease. In T2DM mice, treatment with DAPA attenuated endothelial dysfunction, vascular smooth muscle dysfunction, and arterial stiffness [35]. In their studies, Steven et al. [32] and Gaspari et al. [36] showed that DAPA attenuated TNFα- and hyperglycemia-induced endothelial dysfunction in vitro with a human endothelial cell line. While in vivo, both adult and aged ApoE^−/−^ mice chronically administered with DAPA showed attenuated endothelial dysfunction and less vascular adhesion molecules.

### 2.3. The Role of Metabolic Disturbances in the Development of DCM

Myocardial cells are characterized by their metabolic flexibility, which is the ability to utilize several substrates such as glucose, lactate, and fatty acids (FAs) to generate ATP molecules. In a healthy heart, there is a constant supply of ATP by oxidative phosphorylation of FAs in the mitochondria (60–90%), while maintaining a balance in using other substrates like glucose and lactate [37].

In diseases such as HF and DM, the metabolic balance is impaired. The failing heart increases the utilization of glucose over FAs to increase energy production. However, in a diabetic heart, there may be a metabolic shift toward FA oxidation rather than glucose oxidation. This shift is thought to be due to the chronic hyperglycemia, insulin deficiency, and insulin resistance. The enhanced FA oxidation observed in a diabetic heart might exceed cardiac utilization capacity and predispose the heart to triacylglycerols (TAGs) and ceramides disposition, which in turn contributes to cardiac hypertrophy and stenosis. Along with the burden created by advanced glycated end products (AGEs), the cardiac metabolic changes promote collagen deposition and induce myocardial fibrosis leading to the damage of the cardiomyocytes present in DCM [38].

Furthermore, in cardiac diseases, ischemia and hypoxia promote a shift to anaerobic respiration. The activity of adenosine monophosphate kinase (AMPK), an energy balancing enzyme that promotes anaerobic ATP production, is allosterically regulated by the ratio of AMP to ATP. When ATP is abundant, it binds to AMPK and inactivates it. Therefore, during pathological low energy states when AMP is abundant, AMPK is activated to provide the heart with ATP. In addition to energy production, AMPK activation protects cells against myocardial injury during ischemia, reduces ROS, and attenuates endoplasmic reticulum stress. Additionally, sodium (Na^+^) overload, a characteristic of HF, increases Ca^2+^ efflux which interferes with the Krebs cycle, that is adding up to the metabolic disturbances [39].

## 3. Characteristics of NHE & SGLT Membrane Transporters

### 3.1. NHE Overview

The sodium hydrogen exchangers (NHE) family of integral membrane protein antiporters consists of ten isoforms that function by exchanging sodium cations with protons through cell membranes [40]. NHE1 and NHE3 are two well-studied isoforms involved in renal and cardiovascular homeostasis. NHE1, ubiquitous in mammalian cells, is the dominant isoform in the heart where it regulates intracellular pH, cell volume, and proliferation and shows the highest expression in the kidney [40,41,42]. NHE3 contributes to regulating extracellular volume and BP by the reabsorption of Na^+^ in the kidney [40,43].

#### Activity and Regulation of NHE1 and NHE3

NHE1 influxes Na^+^ in response to intracellular acidification, where the protein exhibits an allosteric binding site for protons. The regulation of NHE1 can also occur in response to different membrane receptors that can exert conformational changes or C-terminal phosphorylation. Extracellular and hormonal pathways, such as angiotensin II (ANG-II), endothelin-I, and thrombin, control the activity of NHE1 regulators. Receptor regulators of NHE1 include protein kinases, G-coupled receptors, and integrin receptors [44]. Tyrosine kinase activation increases NHE1 activity through the Ras-mediated ERK cascade, including Ras downstream effectors such as MEK1/2, Raf-1, and ERK. However, the serine/threonine kinase ERK downstream effector called p90 ribosomal S6 kinase (p90^RSK^) directly phosphorylates NHE1 instead of ERK [45]. The RSK family of transporters includes four isoforms (RSK1-4). RSK1, also designated as p90^RSK^, sustains regular cardiac function, making this enzyme essential [46]. p90^RSK^ hyperactivity induces cardiac hypertrophy and HF. In neonates, p90^RSK^ activation increases c-Fos and Egr-1 expression in ventricular myocytes to promote myocytes’ development [44,47]. Furthermore, p90^RSK^ activation reduces glycogen synthase kinase-3ꞵ (GSK-3β) activity in mice with defective ryanodine receptor ion channels leading to cardiac hypertrophy progression [48]. A study by Takeishi et al. [49] found the aberrant activation of p90^RSK^ in guinea pig pressure overload-induced hypertrophic myocardium. Moreover, patients with dilated cardiomyopathy had higher levels of activated p90^RSK^ than their healthy peers [50,51,52,53,54]. These findings highlight the role of p90^RSK^ in inducing cardiac dysfunction, remodelling, and its role in NHE1 activation.

Akt is another kinase known to regulate NHE1 activity. The duration of Akt activation is the determinant of its effect [50]. Short-term Akt activation promotes physiological hypertrophy during postnatal cardiac development characterized by normal or enhanced contractile function [55], whereas contractile dysfunction characterizes long-term Akt activation [56]. A study done on mouse embryo fibroblasts showed that Akt inhibition reduced NHE1 activity by blocking the translocation of NHE1 to the cell membrane. Furthermore, the upstreaming of Akt enhances p90^RSK^ activation and, thus, plays a role in cardiomyopathy [57,58].

The role of cardiac AMPK, one of the NHE channel regulatory kinases, in cardiac metabolism is not known. However, studies suggest that activating AMPK by phosphorylation triggers the trafficking of glucose transporters (GLUT1 and GLUT4) to the sarcolemma and increases glucose uptake [39,59].

The G protein-coupled receptor subunits Ga_q_ and Ga_13_ also activate NHE1. Ga_13_ activates NHE1 through the GTPase RhoA pathway, whereas Ga_q_ activates NHE1 through the PKC-dependent mechanism. The suppression of PKC in several Ga_q_ protein-coupled receptors, namely α1-adrenergic, vasopressin, and endothelin-1, impairs NHE1 activation. However, in some Ca^2+^ mobilizing Ga_q_-coupled receptors, NHE1 activation can occur independently of PKC. Moreover, integrin receptors can activate NHE1, which may be due to the shared signalling pathway with Ga_13_ that activates NHE1 [60,61].

Other than receptor-mediated regulation, NHE1 regulation occurs through the direct binding of regulatory proteins to the C-terminal. Accessory proteins, which take part in the regulation of cardiac NHE1 activity were also investigated, such as carbonic anhydrase-II (CAII), Ca^2+^-binding proteins (calmodulin and calcineurin B homologous proteins [CHPs]), and phospholipids. Cellular Na^+^ concentration, regulated by NHE1, is instrumental for function, playing roles in Ca^2+^ regulation, metabolism, contractility, and heart stability [62].

Several physiological and hormonal modulators regulate NHE3 activity. The majority of the NHE3 regulatory hormones are coupled to protein kinases associated with intracellular signalling cascades. Different mechanisms such as direct phosphorylation, protein trafficking, and the interaction with accessory proteins modulate NHE3 activity [63,64]. Moreover, in a normal state, the regulation of NHE3 is dependent on its C-terminal phosphorylation. Various kinases, including casein kinase 2 (CK2), serum glucocorticoid-regulated kinase-1 (SGK1), protein kinase A (PKA), Ca^2+^/calmodulin-dependent protein kinase-II (CaMKII), cGKII, GSK-3, AKT, ERK and p90^RSK^ mediate NHE3 phosphorylation [64]. No et al. [65] demonstrated that lysophosphatidic acid (LPA) stimulated NHE3 activity by LPA5 receptor and EGF receptor (EGFR) transactivity. This, in turn, activated proline-rich tyrosine kinase 2 (Pyk2) and ERK specifically in the apical membrane. The authors hypothesized that RSK could be an associated effector of Pyk2 and ERK since RSK is a well-known effector of EGFR and ERK. In contrast, the regulation of RSK by Pyk2 is still not known. The study showed that RSK2, but not RSK1, regulated the direct phosphorylation of NHE3 and concluded that the RSK2 phosphorylation of NHE3 mediates the NHE3 regulation by LPA.

### 3.2. SGLT Receptors Overview

Sodium–glucose co-transporters (SGLTs) are active symporters that belong to the solute-carrier family-5 (SLC5) of active glucose transportation, and facilitate glucose homeostasis [66]. The human SLC5 transporter family contains 12 members, with up to six different SGLT receptors identified in human cells. Functional studies showed that all SLC5 family proteins weigh between 60- to 80-kDa (580–718 amino acids). The most-studied isoforms of this family, SGLT1 and SGLT2, are involved in glucose absorption and glucosuria.

#### Activity and Regulation of SGLT

Several studies have focused on the activity and expression of SGLT under different physiological/pathophysiological settings. SGLT1 expression was in the small intestine, kidneys, liver, lungs, cardiac myocytes, and highly expressed in the human heart. SGLT2 expression was primarily found in the kidney and pancreatic alpha cells [67]. SGLT1 levels are elevated further in cardiac ischemia or hypertrophy disease states. This increase in SGLT1 expression can be linked to the increased phosphorylation of secondary messengers such as ERK 1/2 and the mammalian target of rapamycin (mTOR), involved in the signaling pathways of cardiac ischemia/hypertrophy. However, further studies are required to confirm the proposed mechanism [68].

The kidney plays a vital role in glucose homeostasis by promoting the reabsorption of filtered glucose. The two isoforms carry out reabsorption across the apical cell membranes [69]. SGLT2 is located on the luminal membrane of the proximal convoluted tubule in S1 and S2 segments, whereas SGLT1 is expressed in the S3 segment (Figure 1) [70]. A healthy kidney reabsorbs 90% of filtered glucose from the proximal tubule via SGLT2, whereas a diabetic kidney increases its reabsorption of glucose by 20% more than the normal rate through the overexpression of SGLT2. The active transport of glucose by both isoforms is linked with the transport of Na^+^ into the intracellular fluid [70,71]. The inhibition of this process promotes the reduction in intracellular Na^+^ levels and the excretion of glucose in urine (glucosuria), leading to the correction of hyperglycemia [70].

## 4. Role of SGLT and NHE1 and NHE3 in Diabetes

DM can stimulate the proliferation of vascular smooth muscle cells (VSMCs) to proliferate through insulin and insulin-like growth factor-1 (IGF-1), which is in turn mediated by NHE1. Insulin can stimulate the transcription of NHE1 directly and regulate the activity of NHE1 in nonvascular cells, whereas IGF-1 regulates NHE1 activity in vascular cells. Moreover, hyperglycemia affects the activity of NHE1. For example, hyperglycemia increases the production of diacylglycerol precursors, leading to the PKC activation, consequently activating NHE1. Additionally, NHE1 in VSMCs can be activated by the oxidized LDL which has been shown to be elevated in DM and hyperlipidemia. Furthermore, AGEs react with the extracellular matrix, resulting in the thickening of vessel walls. Moreover, VSMCs adhesion, which is mediated by cell surface integrins and extracellular matrix proteins, promotes PKC activation and the stimulation of NHE1 activity. Interestingly, it was speculated that the glycation of the extracellular matrix protein fibronectin inhibited NHE activity and suppressed the growth of VSMCs [72].

On the other hand, the activity of NHE3 is stimulated as a result of increased levels of insulin, glucose, and specific adipokines in T2DM. The increased activity and upregulation of NHE3 may be instrumental to developing chronic complications in diabetic patients such as diabetic nephropathy and uric acid nephrolithiasis [40]. The early phase of diabetic kidney disease presents changes in eGFR, the elevated reabsorption of salt and water, and expanded extracellular volume, all of which advance to hypertension, hyperfiltration, and eventually renal hypertrophy [73].

The number of main Na^+^ and water transporters are hypothesized to increase in diabetic kidneys as a compensatory mechanism due to extensive water and Na^+^ loss [74]. The study demonstrated that STZ- induced T1DM rats had an increased protein content of Na^+^ and water transporters NHE3 (204% of the vehicle mean), thiazide-sensitive Na^+^/Cl^−^ co-transporter, and α, β, and γ subunits of the epithelial sodium channel. According to another study conducted by Klisic et al. [75] using a similar animal model, brush-border membrane NHE3 activity was significantly higher by 40% after seven days and 37% after 14 days compared to control rats. However, the increased activity of NHE3 was not associated with changes in NHE3 protein or mRNA. Unlike Song et al. [74], they selectively used cortical brush-border membrane vesicle for analysis to reflect proximal tubule NHE3 and not the analysis of whole kidney homogenates. An STZ-induced diabetic rat study by Rasch demonstrated that diabetic kidneys were 67% larger in size, had 22% longer proximal tubules, and 20% longer distal tubules compared to normal rat kidneys [76]. Since the reabsorption of Na^+^ occurs mainly in the proximal tubules, its elongation can easily result in the increased activity of NHE3 [73]. Hyperglycemia also enhances ANG-II production by the stimulation of angiotensinogen and RAAS. This further activates NHE3 via the SGK1 signalling cascade involving phosphatidylinositol 3-kinase (PI3-kinase) and 3-phosphoinositide-dependent protein kinase-1 (PDK1) [77,78]. Another signal cascade of ANG-II induced NHE3 stimulation includes the non-receptor tyrosine kinase (c- Src), PI3-kinase activation, PKC [79,80], and Ca^2+^ and CaMKII [81]. In the proximal tubule, the uptake of albumin requires the involvement of the megalin/cubilin complex. In diabetic nephropathy, there is decreased endocytosis of albumin due to decreased megalin expression, characterized by microalbuminuria [82]. The decreased albumin uptake leads to elevated intratubular albumin concentration, stimulating NHE3 activity and further worsening kidney damage [73]. In the opossum kidney cells, high glucose levels resulted in hypertrophy due to increased osmolality [83]. Consequently, albumin uptake increased because of NHE3 overactivity.

## 5. Role of SGLT & NHE1 and NHE3 in Cardiovascular Diseases

### 5.1. Ischemia-Reperfusion Injury, Cardiac Remodelling, and Hypertrophy

HF is a syndrome often developed after several remodelling processes in the heart that includes LV hypertrophy, fibrosis, and diastolic dysfunction [84]. In diabetes, the heart is in a state of metabolic overload due to cardiac metabolism. Several vital mechanisms were linked to the induction of cardiac impairment and the early development of HF that overlap with other CVDs. NHE and SGLT’s potential relevance to the direct effects in the myocardium will be discussed concerning the early stages of HF development.

As NHE 1 is the main plasma membrane isoform in the heart, it takes an essential part in cardiac functioning in normal and disease states. Hormones such as endothelin-1, ANG-II, and α-adrenergic stimulators, contribute to NHE1 activity in cardiac remodeling [85,86]. The overactivity of NHE1 has been proven to cause several pathological changes in the myocardium, including ischemia-reperfusion injury (IRI), cardiac remodelling, hypertrophy, and apoptosis that eventually can progress to HF. The potential mechanisms underlying the role of NHE1 in the remodelling process can be summarized by the role of both Na^+^ accumulation and mitochondrial remodelling [87]. During the disease state, as an adaptive mechanism, NHE1 activity is increased to correct the reduced intracellular pH. Since the Na^+^/K^+^ ATPase becomes inactive during ischemia, NHE-mediated Na^+^ influx leads to the intracellular accumulation of Na^+^ [87,88,89]. This rise in intracellular Na^+^ consequently leads to the two-fold elevation in intracellular Ca^2+^ by the direct reversal of Na^+^/Ca^2+^ exchanger (Figure 2) [85], leading to an intracellular Ca^2+^ overload which in turn triggers deleterious pathways that lead to myocardial injury, hypertrophy, and subsequent dysfunction (Figure 3). Additionally, the impairment of mitochondrial function and structure due to swelling, ATP depletion/dysfunction, ROS production, and the opening of the mitochondrial permeability transition pore (MPTP) often accompanies cardiac hypertrophy. On the other hand, NHE1 inhibition and gene ablation attenuate the opening of MPTP and balances the amounts of fission and fusion proteins on the mitochondria. Hence, NHE1 inhibition serves as a cardioprotective mechanism to prevent Na^+^ and Ca^2+^ accumulation and the subsequent activation of intracellular pathways, which in turn may improve mitochondrial function and structure integrity, and the cumulative adverse effects on the myocardium [86]. Various studies have shown that the inhibition and genetic ablation of the NHE1 of in vivo models protected the myocardium from ischemia-reperfusion injury [90]. In another study, although transgenic mice models overexpressing NHE1 had no significant effect on cardiac function, intracellular pH, intracellular Na^+^, and ischemia-reperfusion injury, NHE1 inhibition with cariporide prior to the development of ischemia prevented the accumulation of Na^+^ and Ca^2+^ and decreased ischemia-reperfusion injury, showing that baseline NHE1 activity was not the rate-limiting step [89].

The effect of SGLT on diabetic hearts has been well researched within the last few years. Although most of the literature reported on the impact of SGLT2i preventing CVD in T2DM, SGLT2 receptors were not detected in the heart. New data confirmed that SGLT1 expression and activity are upregulated in ischemia, hypertrophic, failing, and diabetic hearts in humans with end-stage cardiomyopathy and animal models [84,91]. Ramratnam et al. [92] reported that the overexpression of SGLT1 in transgenic mice was associated with pathologic cardiac hypertrophy and LV dysfunction. During ischemia, glucose uptake and utilization increase along with a 2-to-3-fold upregulation of SGLT1. This upregulation was postulated to be an adaptive response to injury and as a response to AMPK and ERK1/2 activation [91]. How SGLT1 up-regulation makes an impact is not known, and discrepancies between studies leave it uncertain whether SGLT1 receptors exert a protective or deleterious role in cardiac physiology.

During acute injuries, SGLT1 over-expression facilitates glucose uptake and generates ATP molecules for the heart through anaerobic glycolysis. Kashiwagi et al. [67] provided evidence of the protective role of SGLT1 against IRI. Using the ex vivo murine Langendorff model, they studied the role of SGLT1 inhibition by phlorizin on cardiac function. During IRI, the use of phlorizin resulted in significant impairment in the recovery of LV contractions and increased infarct size (due to increased CPK activity). There was also a reduction in ATP content associated with a decrease in glucose uptake and glycolysis, showing that SGLT1 inhibition during ischemia-reperfusion impairs cardiac metabolism.

**Figure 2 ijms-22-12677-f002:**
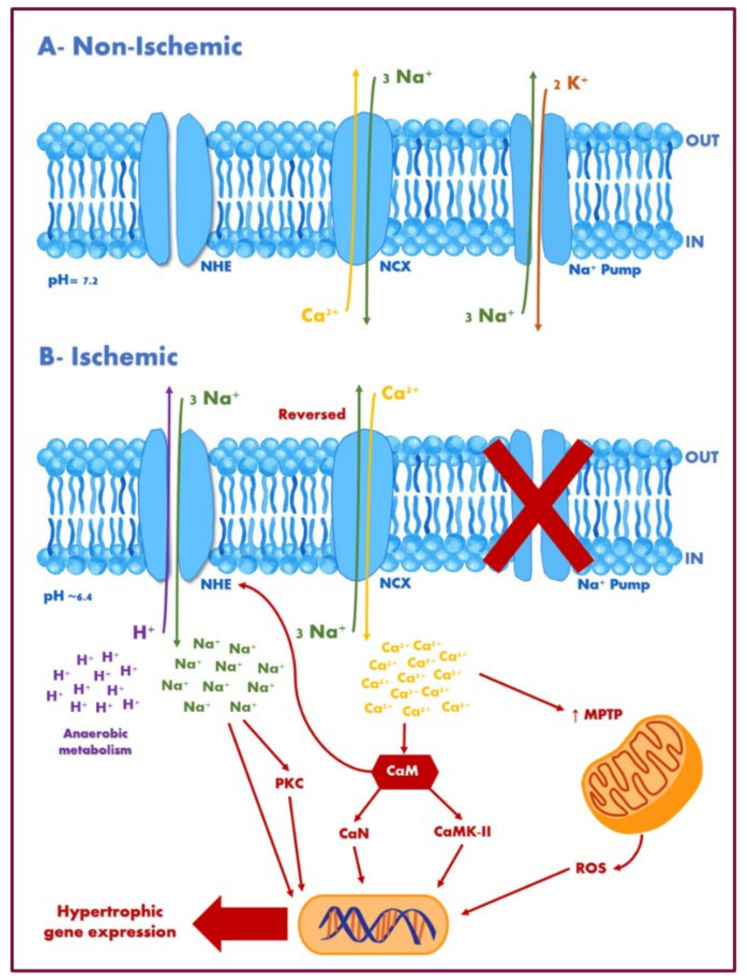
Potential pathways underlying the hypertrophic effect of sodium–hydrogen exchanger 1. (**A**) During non-ischemic events (normal conditions), NHE is relatively quiescent. The Na^+.^ K^+^ ATPase (Na^+^ pump) utilizes ATP to extrude Na^+^, and the bidirectional Na^+^/Ca^2+^ exchanger works predominantly in the forward (Ca^2+^ efflux) mode. (**B**) During ischemic events, intracellular sodium [Na^+^]_i_ rises during ischemia concomitant with a fall in pH. NHE becomes activated in response to intracellular acidosis and other hypertrophic stimulatory factors. Since the Na^+^/K^+^ ATPase becomes inactive during ischemia, NHE-mediated Na^+^ influx leads to the intracellular accumulation of Na^+^. Increased Na^+^ elevates intracellular Ca^2+^ by altering the reversal potential of Na^+^/Ca^2+^ exchangers. Elevated Ca^2+^ activates various pro-hypertrophic factors, including CaN and CaMKII, and increases MPTP, contributing to mitochondrial remodelling. Mitochondrial remodelling results in increased ROS production which, in combination with other factors, contributes to activating transcriptional factors resulting in cardiac hypertrophy. Abbreviations: NHE, sodium–hydrogen exchanger; NCX, Na^+^/Ca^2+^ exchanger; CaM, calmodulin; CaN, calcineurin; CaMKII, Ca^2+^/calmodulin-dependent protein kinase-II; MPTP, mitochondrial permeability transition pore; PKC, protein kinase-C; ROS, reactive oxygen species.

**Figure 3 ijms-22-12677-f003:**
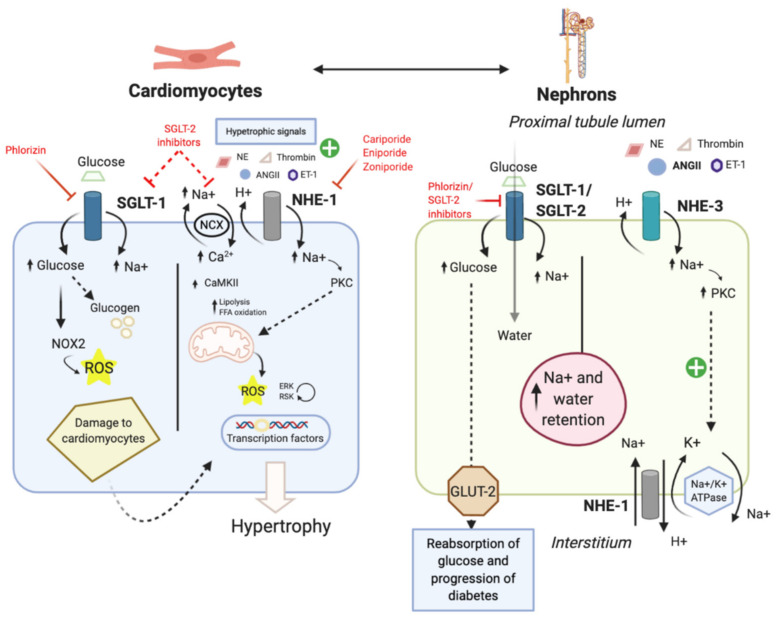
The role of SGLT, NHE, and their inhibitors in diabetes and cardiovascular diseases. Increased SGLT activity in the proximal tubules leads to decreased natriuresis and the increased reabsorption of glucose, and worsening heart failure and diabetes, respectively. In the heart, hypertrophic signals such as endothelin-1, ANG-II, thrombin, and norepinephrine increase NHE1 activity, leading to Na^+^ accumulation and mitochondrial dysfunction which activates the pro-hypertrophic transcription factor. Hyperglycemia leads to increased glucose transport through SGLT1, leading to increased NOX2 activity, and subsequent damage to the cardiomyocytes through ROS. Abbreviations: NOX2, NADPH oxidase; ANG-II, angiotensin II; ET-1, endothelin-1; ERK, extracellular signal-regulated kinase; RSK, ribosomal protein S6 kinase; NE, norepinephrine; GLUT-2, glucose transporter 2; for other abbreviations see Figure 2.

On the other hand, several studies showed that SGLT1 inhibition leads to the improvement of cardiomyopathy. This is evidenced by several experimental studies in SGLT1 knockdown models. Ramratnam et al. [92] reported that double-transgenic mice (SGLT1 knockdown with PRKAG2 mutation) have attenuated cardiac glycogen accumulation, cardiac hypertrophy, and LV dysfunction. Similarly, Li et al. [93] discovered that the pharmacological and genetic inhibition of SGLT1 prevented injuries following ischemia-reperfusion (in in vivo, ex vivo, and in vitro models) and reduced ROS, myocardial necrosis, infarct size, along with improved hemodynamic functions. Furthermore, an in vitro study on adult rat myocytes demonstrated that increased glucose transport through SGLT1 results in NADPH oxidase (NOX2) activation, leading to the increased production of ROS and subsequent damage to cardiomyocytes (Figure 3) [94]. This effect was counteracted by phlorizin, an SGLT1 inhibitor.

In contrast to SGLT1, SGLT2 receptors were not detected in the heart. The mechanism of SGLT2i cardioprotection is still undetermined, but studies have shown that SGLT2i affect cardiomyocytes by directly inhibiting NHE and improving mitochondrial function. Future studies should investigate if there is a link between SGLT2i and SGLT1 in failing hearts and whether dual inhibition may have other beneficial effects on the myocardium. However, Lee et al. [95] induced MI through the left anterior descending artery ligation in non-diabetic mice. Three days after MI induction, there was a transient expression of SGLT2 in the site of occlusion in the heart showed by immunofluorescence and western blot. However, the authors could not conclude if SGLT2 is expressed by cardiomyocytes or by the inflammatory cells migrating to the infarct site.

Overall, there is an interplay between the two membrane transporters (SGLT1 and NHE1) in mediating cardiac effects in failing hearts. Although the role of NHE1 inhibition is well defined, the cardioprotective mechanism of SGLT2 inhibition is unknown with the exemption of SGLT2i direct NHE1 inhibition. Interestingly, a preclinical in vitro study in lipopolysaccharide (LPS)-stimulated mouse cardio-fibroblasts tested the hypothesis that DAPA can cause the NHE1 downregulation of the AMPK-dependent pathway. Ye et al. [96] reported that DAPA resulted in elevated levels of the phosphorylated form of AMPK in LPS-stimulated cardio-fibroblasts. The results showed that DAPA mitigated the rise in NHE1 mRNA and confirmed the relation between NHE1 and Hap70 through the AMPK-dependent pathway. Similarly, Uthman et al. [84], proved that the three SGLT2i available in the market directly suppressed NHE1 activity in vitro.

### 5.2. Diabetic Cardiomyopathy

A plethora of evidence suggests that NHE1 is noticeably involved in mediating cardiac hypertrophic responses in DCM and, therefore, is a potential therapeutic target [97]. Mraiche et al. [98] used two transgenic mouse models: one expressing wild-type NHE1 and another expressing an activated form to investigate the effect of NHE1 activation on cardiac hypertrophy. NHE1 hyperactivation has been linked to elevated glucose levels in DCM induced by PKC-dependent mechanisms. Additionally, an increase in heart weight to body weight, apoptosis, fibrosis, and a decrease in cardiac functionality were recorded.

Studies had shown an enhanced mitochondrial NHE1 activity in the hearts of diabetic rats. Allen and Xiao [99] have illustrated that the main pathway for Na^+^ entry during reperfusion of an ischemic diabetic heart is NHE1. Na^+^ concentration changes are linked to altered Ca^2+^ influx, the production of ROS, and cell damage. Additionally, the expression of the activated form of NHE1 increased the sensitivity to neurohormonal stimulation (using phenylephrine). Indeed, patients with DM experience neurohormonal dysregulation. During HF, neurohormonal systems like norepinephrine, ANG-II, aldosterone, and neprilysin are activated, causing impaired insulin sensitivity and microvascular complications [40,100]. Reduced insulin sensitivity and adipokine abnormalities are characteristic of DM and pathophysiological for HF.

Compared to NHE1, the NHE3 isoform distribution is mainly limited to the kidney and gastrointestinal epithelial cells. The main role of NHE3 in DCM is related to its regulation of Na^+^ reabsorption in the proximal tubules, which regulates sodium uptake following glomerular filtration. NHE3 activity is enhanced with neurohormonal stimulation by norepinephrine, ANG-II, and aldosterone in HF. Additionally, insulin, glucose, and some adipokines, which are elevated in T2DM, stimulate NHE3. In HF, NHE3 activity is elevated in the kidney mediating Na^+^ reabsorption, leading to fluid and Na^+^ retention, peripheral oedema, and diuretic resistance. NHE3 hyperactivity in DM also leads to kidney mesangial cell proliferation, hyperfiltration, and diabetic nephropathy, contributing to cardiac overload and the further worsening of HF [40,101]. Considering all these pathophysiological changes, concluding that the NHE family could link HF and DM is reasonable.

Regarding the SGLT family, evidence shows enhanced SGLT1 expression in end-stage cardiomyopathy in obese mice with T2DM. Controversially, the reduced expression of SGLT1 is recorded in T1DM. This suggests that the increase in its expression might be attributed to the hyperinsulinemia state found in T2DM, but not T1DM. SGLT1 expression was linked to cardiac fibrosis and collagen deposition in the heart [102]. Hypertrophic cardiomyopathy was induced through a transverse aortic constriction in a titin-truncated mouse model that increases interstitial fibrosis in wild-type mice without affecting SGLT1 deficient mice. Additionally, SGLT1 contributes to the oxidative stress seen in DCM, as its destruction in mouse atrium cardiomyocytes protects the cells against hypoxia and reoxygenation injury [103]. Furthermore, mice with cardiomyocyte-specific SGLT1 knockdown were resistant to both in vivo and ex vivo myocardial ischemia/reperfusion injury [104].

SGLT2 is an isoform mainly present in the kidneys, although there is limited-to-no expression in the heart. However, the cardioprotective effects of SGLT2i suggest that SGLT2 is involved in DCM by expression in the kidney, since there is an increased expression of renal SGLT2 and enhanced glucose reabsorption [105]. Studies using knockout mice as a negative control have shown an enhanced SGLT2 expression in T2DM and T1DM mice. However, the biological mechanism for SGLT2 upregulation in DM is not understood. A study with human embryonic cells (HEK-293T) showed that insulin phosphorylated the SGLT2 Ser624 residue, which increased ROS production, further damaging kidney cells [71]. Interestingly, using hypoinsulinemic T1DM, there was also an enhanced expression of SGLT2, which suggests the involvement of other regulatory proteins. ANG-II can increase SGLT2 expression, and its role in inducing cardiac hypertrophy, heart failure, and DCM is proven. This shows a link between the expression of SGLT2 in the kidneys and DCM [105].

### 5.3. Hypertension

Hypertension occurs as an autoregulatory response to increased Na^+^ concentration due to increased reabsorption. Na^+^ reabsorption is mediated by activating the RAAS and the consequent triggering of the ANG-II Type 1 (AT1) receptor, stimulating NHE3-induced Na^+^ influx [43]. Increased Na^+^ influx promotes the significant expansion of extracellular volume and cardiac output and mediates a rise in peripheral vascular resistance resulting in elevated BP [73]. Hypertension signals the body to promote the re-establishment of the expanded volume via decreased eGFR followed by pressure natriuresis.

The overexpression of NHE3 in proximal tubules was detected in the spontaneously hypertensive rat (SHR) model of human primary hypertension [106]. Interestingly, ANG-II leads to the overexpression of NHE3 in the cultured cells of the proximal tubules as it stimulates the exocytosis of NHE3. In fact, it was found that along with NHE regulatory factor 1, IRBIT protein forms a complex with NHE3 during exocytosis after ANG-II stimulation [107]. Other anti-natriuretic peptide hormones such as insulin and glucocorticoid caused the activation of NHE3 in proximal tubules [106,108,109,110]. In studies by Li et al. [107] and [111], the role of NHE3 in hypertension using NHE3^−/−^ mice with the transgenic rescue of NHE3 in the small intestine was investigated, and it affirmed their hypothesis that NHE3 is essential for ANG-II-induced hypertension. In mice with ANG-II-induced hypertension, the selective genetic deletion of NHE3 of the proximal tubule attenuated the condition [106]. Studies showed that 50% of hypertensive individuals were insulin resistant. Moreover, hypertensive patients are at a high risk of developing CVDs [112]. NHE3 participates in Na^+^ reabsorption in proximal tubules and plays a critical role in the absorption of dietary Na^+^ from the gut. Two studies had investigated the role of gut NHE3 using oral NHE3 inhibitor with low systemic absorption on obese SHR. The treatment had significantly reduced the absorption of Na^+^ from the gut and reduced BP [113,114].

NHE1 contributes to pH, salt, and volume regulation, linking it to hypertension. Using NHE1-overexpressing transgenic mice, Kuro-o et al. [115] showed that NHE1 overexpression caused salt-sensitive BP elevation in mice. Primary hypertensive animal models and the peripheral cells of primary hypertensive donors also showed increased NHE1 activation [116]. Conversely, NHE1 knockout in mice leads to a reduction in BP and artery tension [117]. It is suggested that NHE1 overactivity in VSMCs increases intracellular Na^+^, reduces Na^+^/Ca^2+^ exchangers, and leads to elevated intracellular Ca^2+^ and increased contraction. With chronic NHE1 overactivation, abnormal cell proliferation can occur in VSMCs [118]. In proximal tubules, alterations in Na^+^ transporters impact the extracellular volume, thus changing BP independently from transporters in other renal segments. In hypertension, there is an increase in Na^+^ reabsorption that mainly occurs in the proximal tubule and loop of Henle. SGLT2, which is localized in the proximal tubule, is responsible for 60–90% of the renal uptake of Na+ and glucose [119,120].

The relationship between SGLT2 activity and hypertension is not known yet. When [121] compared SGLT2 activity in the proximal tubule of renovascular hypertensive rats with normotensive rats, they saw that Na^+^-dependent glucose uptake and SGLT2 expression were higher in the renovascular hypertensive group. In chronically infused ANG-II Wistar rats, the activity and expression of SGLT2 were increased. Using EMPA did not affect the BP; however, losartan, a RAAS inhibitor, reduced BP. In this study, Losartan prevented renal damage, whereas EMPA produced a minimal protective effect. Nonetheless, EMPA attenuated oxidative stress [122]. Clinical trials have consistently shown that SGLT2i reduces BP [123]. In the EMPA-REG OUTCOME trial, EMPA was correlated with minimal BP reduction [124]. Similarly, CANVAS and CANVAS-R studies showed a reduction in systolic BP by 3.9 mmHg in the canagliflozin (CANA)-treated group compared to placebo [125]. In the DECLARE-TIMI 58 trial, patients treated with DAPA had lower BP by 2.7 mmHg versus placebo [126]. A meta-analysis composed of 27 RCTs with 12,960 participants concluded that SGLT2i resulted in lower systolic and diastolic BP by 4 mmHg (95% CI, −4.4 to −3.5), and 1.6 mmHg (95% CI, −1.9 to −1.3), respectively, from baseline [127].

SGLT2 upregulation could be a partial contributor to hypertension pathogenesis, however several hypotheses explain the role of SGLT2 and its inhibition in BP regulation [123]. Diuresis associated with SGLT2i may cause reduced BP. However, diuresis is a temporary SGLT2i effect, whereas BP reduction from baseline is a sustained effect [123,128].

A direct relationship between SGLT1 and BP has not been established. SGLT1-deficient (SGLT1^−^/^−^) mice show glucose–galactose malabsorption; however, the absence of SGLT1 did not affect BP compared to wild-type mice [129]. BP exhibits a diurnal rhythm and SGLT1 expression exhibits a similar rhythm with the highest expression in the morning [130]. Remarkably, a hypertensive animal model showed a downregulation in SGLT1 function and expression [131]. More research is needed to determine how SGLT1 is involved in hypertension pathophysiology [69].

In summary, SGLT and NHE exhibit different roles in hypertension. Hypertension can worsen the prognosis of DCM where it contributes to the enlargement of the cardiac wall thickness and mass. Increased BP, along with other stimuli, causes vasoconstriction and fluid overload that aggravates cardiac hypertrophy and fibrosis of the myocardium.

## 6. Available Inhibitors and Their Clinical Outcomes

### 6.1. Clinical Evaluation of NHE1 Inhibitors

Substantial evidence supports the protective role of inhibiting NHE1 in reducing IRI development, cardiac hypertrophy, systolic dysfunction, and HF. Several NHE1 inhibitor studies (e.g., cariporide, eniporide, and zoniporide) showed significant protection against CV injuries [40]. Despite that, clinical studies in human subjects showed varying results. Therefore, a cardioprotective role of NHE1 inhibition in humans is controversial.

The ESCAMI randomized trial investigated eniporide effect on patients (n = 1389) with ST-elevation MI for the primary outcome of the change in infarct size with eniporide as add-on therapy to reperfusion in IRI [132]. However, eniporide did not reduce the infarct size nor improve patients’ clinical outcomes. However, the protective effect of cariporide in animal models may have been due to the administration of cariporide during ischemia and not during reperfusion [133]. Additionally, Klein etl al. [133] and Rupprecht et al. [134] tested the effect of cariporide (40 mg) on 100 patients with acute anterior MI getting direct coronary angioplasty. Compared to placebo, patients who received cariporide had higher ejection fraction (50% vs. 40%; *p* < 0.05), lower end-systolic volume (69 vs. 97 mL; *p* < 0.05), significant improvement in wall motion abnormalities, and a reduced cumulative release of CK-MB (*p* = 0.047). Thus, NHE inhibition by cariporide may prevent reperfusion injury and aid in the recovery from ventricular dysfunction. This study contradicts the ESCAMI study’s findings concerning the effects of NHE inhibition, as an adjunct to reperfusion therapy, on the myocardium. The GUARDIAN study assessed the safety and efficacy of cariporide (20, 80, or 120 mg) in a cohort of patients (n = 11,590) at risk for myocardial necrosis [135]. The cardioprotective effect was only evident in patients who underwent coronary artery bypass graft surgery (CABG) and treated with 120 mg cariporide. The EXPEDITION study was the first phase 3 myocardial protection trial to examine cardioprotective effects of cariporide in high-risk patients (n = 5761) undergoing CABG [136]. The drug resulted in increased mortality rates associated with increased cerebrovascular events (2.2% with cariporide vs. 1.5% with placebo; *p* = 0.02). The incidence of death or MI was significantly reduced from 20.3% in the placebo group to 16.6% in the cariporide group (*p* = 0.0002). However, due to the increased mortality, the study was terminated early. The findings suggested that NHE1 inhibition could significantly reduce ischemia-reperfusion injuries and that cariporide is unlikely to be used clinically. The mixed findings obtained from the clinical research of NHE inhibitors conflict with the highly favourable evidence from experimental studies and emphasize the challenges facing the translation of potential therapies from the laboratory to the clinic.

### 6.2. Clinical Evaluation of SGLT Inhibitors

As SGLT1 and SGLT2 are considered the primary transporters involved in glucose homeostasis, several drugs have been developed to inhibit their activity. Inhibiting SGLT1 results in better post-meal blood glucose control by blocking glucose uptake in the intestine, which decreases the glycemic burden. Furthermore, as most glucose reabsorption processes in the proximal convoluted tubule are mediated by SGLT2, the inhibition of this transporter reduces the kidney glucose threshold, and the excretion of glucose lowers glucose plasma levels. This effect is insulin-independent, and therefore, if this class of inhibitor is used alone, the risk of hypoglycemia is low. These drugs can also increase weight loss by promoting urinary glucose excretion [137].

The development of SGLT inhibitors started in 1835 with the discovery of phlorizin, which was speculated to treat malaria and infections until 1886 when it was reported to cause glucosuria and renal effects [138,139]. The administration of subcutaneous phlorizin to diabetic rats with insulin resistance normalized insulin sensitivity and glucose levels [140,141]. However, the clinical use of phlorizin was limited due to its poor bioavailability, low solubility, and non-selectivity in SGLT inhibition with increased selectivity to SGLT2 compared to SGLT1 [138,142].

Due to the limitations of phlorizin, other compounds were developed, such as T-1095 and its active form T-1095A, which are synthetic compounds derived from phlorizin. Oral T-1095 exhibited dose-dependent elevation in urine glucose excretion by inhibiting SGLT2 in the proximal tubule, resulting in a reduced blood glucose concentration [143]. Additionally, T-1095 reduced postprandial blood glucose levels in STZ-induced diabetic rats via the inhibition of SGLT1 in the intestine. However, the clinical use of T-1095 was limited due to its non-selectivity.

Currently, several SGLT2 inhibitors are approved for clinical use in the US and worldwide, and others are under development. Sotagliflozin is an example of a dual SGLT1/2 inhibitor with only a ~30-fold higher selectivity for SGLT2 over SGLT1, and researchers are seeking its approval by the FDA [138]. Two randomized controlled trials, SOLOIST-WHF and SCORED, randomized T2DM patients with CKD or recent HF hospitalizations, respectively, to receive either sotagliflozin or placebo, and found a statistically significant reduction in death from cardiovascular causes, HF hospitalizations, urgent visits for HF, and all-cause mortality [144,145]. Moreover, sotagliflozin was shown to decrease HbA1c and fasting blood glucose in a dose-dependent manner by providing better glycemic control with no increase in hypoglycemic effects [146,147]. Sands et al. conducted a randomized, multicenter, placebo-controlled, double-blind trial to assess the safety and efficacy of Sotagliflozin, as a combined therapy with insulin in 33 T1DM patients. The results showed a 32.1% reduction from baseline in the total daily bolus insulin dose in the Sotagliflozin-treated group compared to a 6.4% reduction in the placebo group (*p* = 0.007), and a baseline reduction in HbA1c by 0.55% in the treatment group compared to 0.06% in the placebo group (*p* = 0.002). Additionally, it was reported that the treatment group mean body weight was reduced by 1.7 kg compared to a gain in weight by 0.5 kg in the placebo group (*p* = 0.005) [148].

In Tandem3, a phase 3 double-blind clinical trial, the authors concluded that the proportion of patients with T1DM on insulin and who received sotagliflozin accomplished a HbA1c level of less than 7.0% (without severe hypoglycemia or diabetic ketoacidosis (DKA)) was larger than the placebo group. However, the sotagliflozin group had a higher rate of DKA than the placebo group [149]. Other SGLT inhibitors are still under investigation, such as mizagliflozin, a selective SGLT1 inhibitor, and licogliflozin, a dual SGLT1/2 inhibitor [138].

Recently, several SGLT2i were developed and approved to be used in T2DM patients. In addition to their glucose-lowering effects, CANA, DAPA and EMPA showed clinical evidence of improved clinical outcomes of HF, chronic kidney disease, and CVD in patients with adequate eGFR. [91]. CANA, at higher concentrations (300 mg), was shown to exhibits a dual effect by inhibiting SGLT-2, causing increased glucose excretion and delayed intestinal glucose absorption via SGLT-1 inhibition [150]. The CANVAS program joined the analysis of CANVAS and CANVAS-R, which included patients with T2DM and increased CV risk to assess CANA use compared to placebo [125]. The CANVAS trial assessed CV risk and major adverse cardiac events, whereas the CANVAS-R trial assessed the progression of albuminuria in patients using CANA versus placebo. The combined analysis showed CANA lowers CV events and probably attenuates albuminuria progression and suppressed the decline of the glomerular filtration rate (eGFR). However, it increases the risk of metatarsal amputation compared to placebo. EMPA is another example of an SGLT2 inhibitor with cardioprotective evidence. In the EMPA-REG OUTCOME trial [124], EMPA was reported to reduce CV death by 38%, HF hospitalization by 35%, and death from any cause by 32% in T2DM patients at high CV risk. The trial showed that EMPA reduced worsening nephropathy and progression to macroalbuminuria compared to placebo. Additionally, the DECLARE-TIMI trial evaluated the effect of DAPA in patients with T2DM and established CVD or CV risk factors [126]. Although DAPA was associated with lower rates of HF hospitalization or CV death than placebo, there was no difference in major adverse cardiac events between placebo and DAPA. Furthermore, the DAPA-HF and EMPEROR-REDUCED trials found a protective effect of DAPA and EMPA, respectively, against CV death and HF hospitalizations in HF patients regardless of the presence of diabetes [151,152]. Interestingly, a preclinical in vitro study in lipopolysaccharide (LPS)-stimulated mouse cardiofibroblasts, tested the hypothesis that DAPA can cause NHE1 downregulation via the AMPK-dependent pathway. Ye et al. reported in the study that DAPA resulted in elevated levels of the phosphorylated form of AMPK (P-AMPK) in LPS stimulated cardiofibroblasts. Moreover, the results show that DAPA attenuated the rise in NHE1 mRNA and the relation between NHE1 and Hap70 through the AMPK dependent pathway [96]. Similarly, in a recent study by Uthman et al., the use of SGLT2 inhibitors DAPA, EMPA, and CANA directly suppressed NHE1 activity in vitro [84].

Other clinical studies pointed to the natriuretic effects of SGLT2i, which impact CV benefits through a reduction in fluid retention and the risk of developing HF. Using immunofluorescence, Pessoa et al. [153] reported that NHE3 co-localizes with SGLT2, not SGLT1, concluding that SGLT2i causes diuresis via NHE3 inhibition. A recent randomized placebo-controlled crossover study in 20 patients with T2DM and HF treated with EMPA monotherapy showed a significant increase in the fractional excretion of Na^+^ (FENa) compared to placebo (*p* = 0.001). A synergistic effect on the FENa was reported when combined with bumetanide (*p* = 0.001). Moreover, after 14 days of SGLT2 inhibition by EMPA and its persistent natriuretic effect, there was a reduction in blood volume (*p* = 0.035) and plasma volume (*p* = 0.04) without inducing neurohormonal activation, off-target electrolyte wasting, and renal dysfunction. Thus, the benefits of the long-term use of EMPA in HF patients may be volume management that is attributed to natriuretic effects [154].

Another proposed mechanism stated that EMPA projects cardiac benefits by restoring the calcium concentration in the mitochondria of the myocytes. The NHE1-mediated influx of sodium causes the efflux of calcium through the NCX, thereby, decreasing its levels in the mitochondria causing impairment, energy depletion, and oxidative stress. Imbalance in calcium homeostasis can also alter the contraction–excitation integration and lower contractility, observed in DCM and HF [155,156]. NHE inhibition would restore depleted calcium levels and decrease the risk of cardiac damage. As per these findings, the American Diabetes Association recommends a combination therapy of metformin and a SGLT2 inhibitor for established ASCVD, HF, or chronic kidney disease [157].

To sum up, there is substantial proof that SGLT2 inhibitors offer cardioprotective and renoprotective effects that are considerably more than would be predicted based on their effects on glycemia or glycosuria. These cardiorenal benefits cannot be solely explained by the effect of SGLT-2i in lowering blood glucose or the natriuretic action. Recent studies hypothesized that SGLT2 inhibitors’ ability to stimulate ketogenesis accounts for their beneficial effects on the heart and kidney because increased ketone body synthesis may potentially offer an efficient fuel that could augment the energy status of stressed organs. Moreover, the stressed heart favorably utilizes ketone bodies and the diabetic kidney is a ketogenic organ. Thus, ketogenesis is postulated to contribute to the development of diabetic nephropathy. It is more favorable to say that SGLT-2i works by mimicking the starvation state by stimulating the activity of AMPK and suppressing Akt/mTOR signaling to reduce oxidative stress, reduce inflammation, normalize mitochondrial function and structure, prevent IRI, and prevent the development of cardiomyopathy [158,159].

## 7. Future Directions and Perspectives

Diabetes mellitus is highly associated with cardiovascular disease, as hyperglycemia triggers cardiac metabolic imbalances, endothelial dysfunction, ROS production, RAAS activation, and impaired Ca^2+^ homeostasis, leading to heart failure. There is increasing evidence supporting the cardioprotective role of SGLT2i, in which the mechanism by which this occurs remains unclear. Overall, there are numerous studies that support the role that the crosstalk between SGLT and NHE could have in contributing to the cardiac effects seen in failing hearts. NHE1 and NHE3 are two well-studied isoforms involved in renal and cardiovascular homeostasis. In the heart, NHE1 regulates intracellular pH, cell volume, proliferation, and Na^+^ concentration, which in turn plays a role in Ca^2+^ regulation, metabolism, contractility, and the stability of the heart. On the other hand, renal NHE3 contributes to the regulation of extracellular volume and BP. Although the role of NHE1 inhibition is well defined, the exact cardioprotective mechanism of SGLT2 inhibition has not been determined, with the exception of SGLT2i-directed NHE1 inhibition. Further studies are needed to investigate the interaction between NHE3 and SGLT2. Additionally, the cellular interplay between NHE1 and NHE3 in cardiometabolic diseases needs to be further investigated.

## Figures and Tables

**Figure 1 ijms-22-12677-f001:**
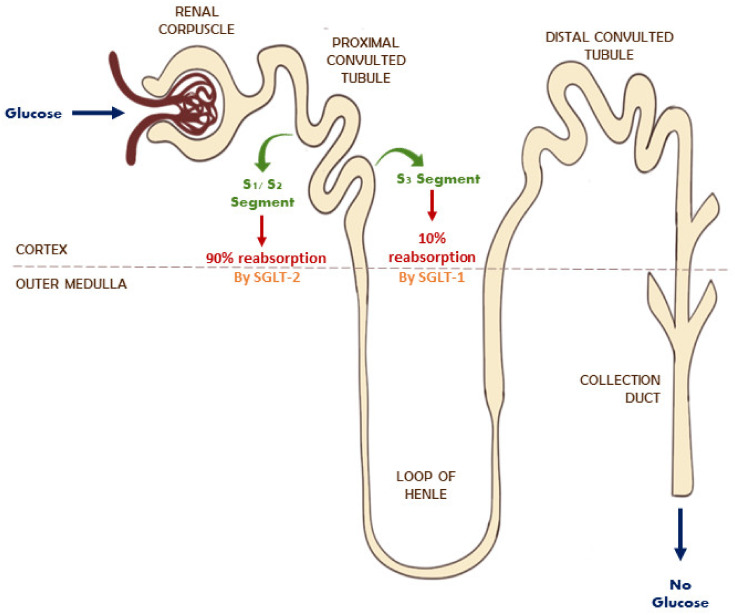
Glucose reabsorption through SGLT1 & SGLT2 in the normal kidney.

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
