# Peer review of "Crosstalk between Sodium–Glucose Cotransporter Inhibitors and Sodium–Hydrogen Exchanger 1 and 3 in Cardiometabolic Diseases"

_ijms, 2021, doi:10.3390/ijms222312677_

Round 1
Reviewer 1 Report
The present review manuscript by Al-Shamasi et al. describes the role of sodium-glucose cotransporter inhibitors and sodium-hydrogen exchanger-1 and 3 in cardiometabolic diseases. The incomplete current understanding of the molecular patho-mechanism justifies the objective of the review manuscript in hand.
The paper is generally well written, and the topic is of interest. However, there are several issues that need to be discussed.
The following comments are for the author’s consideration:
- Include fibrogenesis processes in Heart and kidneys during diabetes and cardiometabolic diseases.
- Mechanisms of fibrosis and their regulators during cardiometabolic diseases.
- Discuss more widely the connections among diabetic nephropathy, cardiovascular dysfunctions and cardiometabolic disorders. That I find the one of the objectives of this manuscript.
- Describe the key molecules such as endothelial FGFR1 regulates fibrosis in diabetic heart and kidneys. Describe the key molecules and regulator that are important for endothelial cell health in diabetes such as SIRT3, glucocorticoid receptors etc. EndMT is the one the processes for the generation of myofibroblasts in heart and kidneys.
- How does alteration in metabolic flexibility affect the cardiometabolic diseases during diabetes. Myofibroblasts metabolic shifts in heart and kidneys
- Different class of SGLT-2i in organ protection. Their mechanism such as empa restores normal kidney phenotype by inhibiting abnormal defective glucose metabolism linked EMT in kidneys.
- Discuss the different class of SGTLs inhibitors and NHE-1 inhibitors and their effect on the blood glucose levels and on hemodynamic alterations
- Future directions and perspectives
Author Response
Dear Respected Reviewer, thank you for your recent review of the review article entitled "Crosstalk Between Sodium-Glucose Cotransporter Inhibitors and Sodium-Hydrogen Exchanger- 1 and 3 in Cardiometabolic Diseases" (ijms-1423943). We respectfully submit to you the revised version of the manuscript as per your kind feedback for your consideration. We have outlined our responses in point form below.
We thank you in advance for reviewing this article and look forward to hearing back from you.
Sincerely,
Dr. Fatima Mraiche
- Include fibrogenesis processes in the Heart and kidneys during diabetes and cardiometabolic diseases. >> Thank you for this great suggestion. We have elaborated on this point in section 2.1 of the review article.
- Mechanisms of fibrosis and their regulators during cardiometabolic diseases. >> Thank you for this great suggestion. We have elaborated on this point in section 2.1 of the review article.
- Discuss more widely the connections among diabetic nephropathy, cardiovascular dysfunctions, and cardiometabolic disorders. That I find one of the objectives of this manuscript. >> Thank you for your feedback. We cover this in section 4 of the review article.
- Describe the key molecules such as endothelial FGFR1 regulates fibrosis in diabetic heart and kidneys. Describe the key molecules and regulator that are important for endothelial cell health in diabetes such as SIRT3, glucocorticoid receptors etc. EndMT is the one the processes for the generation of myofibroblasts in heart and kidneys. >> Thank you for this great suggestion. We have elaborated on this point in section 2.1 and 2.2 of the review article.
- How does alteration in metabolic flexibility affect the cardiometabolic diseases during diabetes. Myofibroblasts metabolic shifts in heart and kidneys >> >> Thank you for this great suggestion. We have elaborated on this point in section 2.3 of the review article.
- Different class of SGLT-2i in organ protection. Their mechanism such as empa restores normal kidney phenotype by inhibiting abnormal defective glucose metabolism linked EMT in kidneys. >> Thank you for this great suggestion. We have elaborated on this point in section 6.2 of the review article.
- Discuss the different class of SGTLs inhibitors and NHE-1 inhibitors and their effect on the blood glucose levels and on hemodynamic alterations >> Thank you for your feedback. We cover this in section 4 of the review article.
- Future directions and perspectives>> Thank you for yoru feedback. We have revised the title of section 7 and entitled it future directions and perspectives and have revised the paragraph to be more reflective of the section title.
Reviewer 2 Report
In this review article, the authors outlined the possible mechanisms predisposing to diabetic cardiomyopathy and discussed the interaction between sodium hydrogen exchangers (NHE) and sodium-glucose cotransporter-2 inhibitors (SGLT2i) in cardiovascular disease.
Comments
This is an interesting review article. This manuscript is well-written. The reviewer has some minor concerns as follows:
- The format in the text for the first author cited from the references can be changed. For examples, line 153, “Gaspari, Spizzo [30] showed …” changes to “Gaspari et al. [30] showed…”; line 209, “A study by Takeishi, Abe [43] found…” changes to “A study by Takeishi et al. [43] found…”; and others. There are many sentences in the text having these inappropriate presentations that can be revised.
- In Figures 2 and 3, the full names for abbreviations in the figures can be described in the legends.
- In the text, there are typing errors and inappropriate words that need to be carefully checked and corrected. For examples, line 78, “…stage. [9].” changes to “…stage [9].”; line 264, “cells. [61].” changes to “cells [61].”; line 315, “…compared to normal rat kidneys Rasch [70].” changes to “compared to normal rat kidneys [70].”; lines 368-369, for Na+ or K+, “+” needs to be superscript; and others.
Author Response
Dear Respected Reviewer, thank you for your recent review of the review article entitled "Crosstalk Between Sodium-Glucose Cotransporter Inhibitors and Sodium-Hydrogen Exchanger- 1 and 3 in Cardiometabolic Diseases" (ijms-1423943). We respectfully submit to you the revised version of the manuscript as per your kind feedback for your consideration. We have outlined our responses in point form below.
We thank you in advance for reviewing this article and look forward to hearing back from you.
Sincerely,
Dr. Fatima Mraiche
Response to Reviewer:
- The format in the text for the first author cited from the references can be changed. For examples, line 153, “Gaspari, Spizzo [30] showed …” changes to “Gaspari et al. [30] showed…”; line 209, “A study by Takeishi, Abe [43] found…” changes to “A study by Takeishi et al. [43] found…”; and others. There are many sentences in the text having these inappropriate presentations that can be revised. >> Thank you for your thorough review. Kindly note that we have revised the review article as suggested.
- In Figures 2 and 3, the full names for abbreviations in the figures can be described in the legends. >> Thank you for your thorough review. Kindly note that we have revised the review article as suggested.
- In the text, there are typing errors and inappropriate words that need to be carefully checked and corrected. For examples, line 78, “…stage. [9].” changes to “…stage [9].”; line 264, “cells. [61].” changes to “cells [61].”; line 315, “…compared to normal rat kidneys Rasch [70].” changes to “compared to normal rat kidneys [70].”; lines 368-369, for Na+ or K+, “+” needs to be superscript; and others. >> Thank you for your thorough review. Kindly note that we have revised the review article as suggested.
Round 2
Reviewer 1 Report
Please make sure all references have been appropriately placed.